# Potential of Coccolithophore Microalgae as Fillers in Starch-Based Films for Active and Sustainable Food Packaging

**DOI:** 10.3390/foods12030513

**Published:** 2023-01-23

**Authors:** Ana S. P. Moreira, Joana Gonçalves, Francisco Sousa, Inês Maia, Hugo Pereira, Joana Silva, Manuel A. Coimbra, Paula Ferreira, Cláudia Nunes

**Affiliations:** 1Department of Chemistry, CICECO—Aveiro Institute of Materials, University of Aveiro, Campus Universitário de Santiago, 3810-193 Aveiro, Portugal; 2LAQV-REQUIMTE—Associated Laboratory for Green Chemistry of the Network of Chemistry and Technology, Department of Chemistry, University of Aveiro, Campus Universitário de Santiago, 3810-193 Aveiro, Portugal; 3Department of Materials and Ceramic Engineering, CICECO, University of Aveiro, Campus Universitário de Santiago, 3810-193 Aveiro, Portugal; 4CCMAR—Centre of Marine Sciences, University of Algarve, Campus de Gambelas, 8005-139 Faro, Portugal; 5GreenCoLab—Associação Oceano Verde, University of Algarve, Campus de Gambelas, 8005-139 Faro, Portugal

**Keywords:** bioplastics, *Emiliania huxleyi*, *Chrysotila pseudoroscoffensis*, polysaccharides, starch, calcium carbonate, antioxidant activity

## Abstract

Coccolithophore microalgae, such as *Emiliania huxleyi* (EHUX) and *Chrysotila pseudoroscoffensis* (CP), are composed of calcium carbonate (CaCO_3_) and contain bioactive compounds that can be explored to produce sustainable food packaging. In this study, for the first time, these microalgae were incorporated as fillers in starch-based films, envisioning the development of biodegradable and bioactive materials for food packaging applications. The films were obtained by solvent casting using different proportions of the filler (2.5, 5, 10, and 20%, *w*/*w*). For comparison, commercial CaCO_3_, used as filler in the plastic industry, was also tested. The incorporation of CaCO_3_ and microalgae (EHUX or CP) made the films significantly less rigid, decreasing Young’s modulus up to 4.7-fold. Moreover, the incorporation of microalgae hydrophobic compounds as lipids turned the surface hydrophobic (water contact angles > 90°). Contrary to what was observed with commercial CaCO_3_, the films prepared with microalgae exhibited antioxidant activity, increasing from 0.9% (control) up to 60.4% (EHUX 20%) of ABTS radical inhibition. Overall, the introduction of microalgae biomass improved hydrophobicity and antioxidant capacity of starch-based films. These findings should be considered for further research using coccolithophores to produce active and sustainable food packaging material.

## 1. Introduction

Food packaging has an important role in protecting food from degradation during distribution and storage, minimising food waste. However, fossil-based plastics are widely used as food packaging materials, with several environmental concerns related to their non-renewable origin and non-biodegradable nature [1,2,3]. In this context, also driven by European Union’s plastic strategy for a circular economy [1], bioplastics have emerged as environmentally friendly alternatives for reducing the use and waste of conventional plastics [3]. Particularly, biodegradable films have been developed using matrix polysaccharides from natural renewable sources, envisioning their application as packaging in the food industry [4,5,6,7].

Regarding polysaccharide-based films, several studies have been performed using starch, considering its high abundance in nature as wastes and, consequently, low cost, and capacity to become thermoplastic. Nevertheless, starch-based films typically have lower mechanical and gas barrier properties than synthetic polymers [2,8], which are important characteristics for food packaging. To improve such properties, several works have explored the reinforcement of starch matrix with fillers or additives, either of organic (e.g., cellulose fibres) or inorganic (e.g., calcium carbonate) nature [2,9,10]. In particular, the addition of CaCO_3_ nanoparticles (0.02–0.5%) to corn starch films, prepared using glycerol as a plasticiser and by solvent casting technique, increases tensile strength and elongation at break and Young’s modulus, while water vapour permeability (WVP) decreases [10]. The same effect on mechanical properties is observed by the addition of CaCO_3_ nanoparticles up to 1% into polycaprolactone/chitosan composites prepared by melt-mixing and hot compression. In contrast, for higher amounts (3, 5, or 7%), a reduction of tensile strength and elongation at the break is observed, which can be associated with the bad dispersion and agglomeration of nanoparticles [11]. Also noteworthy is the development of active starch-based films by the incorporation of bioactive compounds, namely with antioxidant capacity, to protect foodstuffs from oxidation reactions and extend their shelf-life [2,12].

Microalgae contain several bioactive molecules, such as polysaccharides, proteins, lipids (rich in omega-3 fatty acids), and pigments [13,14,15]. Indeed, microalgae have increasing interest as sustainable healthy foods, but also as promising for development of new products and materials [13,14,16,17,18], including bioplastics [18]. For example, previous studies reported the use of microalgae biomass in the formulation of starch-based films, namely glycerol-plasticised films obtained by solvent casting [19,20,21]. In such studies, the general trend when incorporating microalgae biomass, namely *Nannochloropsis gaditana* (2%) [19] and *Herochlorella luteoviridis* or *Dunaliella tertiolecta* (0.5, 1, and 2%) [20], was the decrease in tensile strength (indicative of less mechanically resistant films). Water vapor permeability also decreases [19,20], except when using *H. luteoviridis* at 0.5% and 2% [20]. An increase in elongation at break (indicative of more flexible films) was observed with the highest amount of microalgae biomass (2%) [19,20]. Improved antioxidant activity was reported by addition of *H. luteoviridis* or *D. tertiolecta* to starch-based films containing glycerol as plasticizer [20], as well as of *Tetradesmus obliquus* to starch-based films containing glycerol plus polyallylamine (the last used as anti-plasticisation agent) [21]. Antioxidant components of the microalgae biomass, such as pigments, are related to the enhanced antioxidant potential of the films [20,21]. Despite these types of films still not being applied in commercialised foods, films with *H. luteoviridis* extract were tested in salmon packaging, inhibiting lipid oxidation (but not avoiding moisture loss) [20].

Coccolithophore microalgae (classified in phylum Haptophyta and class Coccolithophyceae) have in their composition both bioactive compounds (such as lipids [22,23] and pigments [24]) and CaCO_3_ plates that can be explored to produce sustainable active packaging. As a characteristic of this class, the CaCO_3_ plates, known as coccoliths, form an external covering of the cell surface, a coccosphere. Coccoliths have unique and extremely sophisticated structures (not found in synthetic compounds), with different shapes and size among coccolithophore species [25,26,27]. The unique morphological characteristics of coccoliths make them appealing for material applications. For example, coccoliths exhibit a higher surface area than typical commercialised synthetic calcite particles [28]. However, to the best of our knowledge, there is no published study exploring the potential of coccolithophores as source of valuable compounds. Most studies on coccolithophores have been performed with *Emiliania huxleyi* (an abundant and widely distributed species in the ocean) and focused in their role on the global ocean calcification and carbon cycle (e.g., [29]), or in their unique coccolith structures (e.g., [26]). Indeed, little focus has been paid to the potential of coccolithophores as a raw material, or source of selected valuable compounds, for development of new products and materials. Such studies, along with the sustainable cultivation of these microalgae on an industrial scale to ensure biomass stocks with a constant composition for commercial uses, are needed to boost the economic value of coccolithophore microalgae.

Regarding the chemical composition of microalgae, it is well-known that it depends on growing conditions (e.g., light, temperature, and nutrients), but it is also species-specific [13]. A previous study with cultured *E. huxleyi* biomass revealed 50.8% ash, 20.3% lipids, 15.3% proteins, and 3.7% sugars [22]. Using the same methodologies for characterisation of other coccolithophore species, *Chrysotila pseudoroscoffensis* (originally named as *Pleurochrysis pseudoroscoffensis*) showed 45.5% ash, 11.6% proteins, 11.0% carbohydrates, and 6.4% lipids [23].

Considering their distinct chemical composition, in this study *E. huxleyi* and *C. pseudoroscoffensis* biomasses were incorporated as fillers in starch-based films, envisioning the development of biodegradable and bioactive materials for food packaging applications. The present work is the first one using coccolithophore microalgae for such purpose, particularly in the development of starch-based films. The films were produced by solvent casting, and the microalgae biomass effect on the morphological, mechanical, barrier, and antioxidant properties was evaluated. Commercial CaCO_3_ (calcite), a widely used filler in the plastic industry [30], was used for comparison purposes.

## 2. Materials and Methods

### 2.1. Materials

Potato starch and anhydrous calcium chloride were purchased from Sigma Aldrich (St Louis, MO, USA). Glycerol was purchased from Fisher Chemicals (98%), calcium carbonate from Merck (Darmstadt, Germany) and 2,2′-azino-bis(3-ethylbenzothiazoline-6-sulfonic acid) (ABTS) from Fluka (Buchs, Switzerland). Sodium azide was distributed from Panreac Quimica SAU (Barcelona, Spain). All the used reagents were of analytical grade.

### 2.2. Microalgae Biomass

The strain of *E. huxleyi* (AC453) was obtained from Roscoff Culture Collection (RCC1250), while the strain of *C. pseudoroscoffensis* (ES-PL0118-01) was provided by Instituto de Ciencias Marinas de Andalucía (ICMAN). Both microalgae were cultured under conditions previously described [22,23], as presented below. For both cultures, growth was monitored every day by optical density (750 nm) and microscopic observations. When cultures reached the late exponential growth phase, the biomass was collected by centrifugation (1735× *g* for 15–20 min) and freeze-dried (LyoQuest Telstar).

#### 2.2.1. Culture of *E. huxleyi*

The starter inoculum of *E. huxleyi* was grown in a climatic chamber (20 °C with 12:12h light:dark cycle, irradiance of 50 µmol m^−2^ s^−1^) using 50 mL Erlenmeyer flasks and modified K/2 medium [31]. The concentrated inoculum was then scaled up to 500 mL and 5 L round flasks supplemented with NaHCO_3_ (0.087 g L^−1^) and modified ALGAL medium [32] to a final nitrate concentration of 0.4 mM. All cultures were grown at the standard room temperature of the laboratory (22 ± 2 °C) under natural light without aeration, for about 15 days. All flasks were manually agitated twice a day to keep the cells in suspension.

#### 2.2.2. Culture of *C. pseudoroscoffensis*

The starter inoculum of *C. pseudoroscoffensis* was grown in 25 and 50 mL Erlenmeyer flasks using modified ALGAL medium [32] at a nitrate concentration of 4 mM and a photosynthetic photon flux density (PPFD) of 50 µmol m^−2^ s^−1^ (fluorescent lamps) at ambient temperature (22 ± 3 °C). The concentrated inoculum was progressively scaled-up to higher volumes of 100, 500, and 1000 mL, using the previously described growth conditions. During this stage, cultures were shaken manually, two times a day, to prevent sedimentation and improve the photosynthetic efficiency. Thereafter, several 1 L concentrated inocula were used to inoculate the 5 L bottles used to produce the biomass. Cultures were grown using the same culture medium and temperature, under continuous light (L:D, 24:0), but the PPDF was increased to 100 µmol m^−2^ s^−1^. At this stage, a gentle aeration was introduced in the production vessels. The introduction of air in the production system was attainable due to a process of acclimatisation by gradually increasing the flux of air during growth.

### 2.3. Elemental Analysis of Microalgae Biomass

Microalgae biomass (2 mg, in triplicate) was analysed by elemental analysis (C, H, N and S) on a Leco Truspec-Micro CHNS 630-200-200 elemental analyser at combustion furnace temperature 1075 °C and afterburner temperature 850 °C. Nitrogen was detected using thermal conductivity. Nitrogen-to-protein conversion factor of 4.78 was used to calculate the protein content, as previously proposed for microalgae [33].

### 2.4. Production of Starch-Based Films

For each film, commercial potato starch was used, with different proportions of the microalgae biomass related to the starch matrix (2.5, 5, 10 and 20 % *w*/*w*). To have a homogeneous biomass for film production, freeze-dried microalgae were previously sieved (AS 200 control, Retsch, Germany) using a sieve with 75 µm pore size. Sieved microalgae biomass was suspended in 50 mL of distilled water (1.5% *w*/*v* related to the starch dry weight) and kept under stirring for 1 h. The starch (750 mg) was then added and, after 15 min of stirring, the glycerol (225 mg, corresponding to 30% *w*/*w* related to the starch dry weight) was added. For starch gelatinisation, the mixture was left in a water bath at 95 °C for 30 min under stirring. Afterwards, the solution was degassed under vacuum to remove air bubbles. An amount of 21 g was transferred to plexiglass plates (144 cm^2^ area and 3 mm deep). The plates were then placed in an oven with air circulation at 25 °C for 16 h, resulting films produced by solvent casting. For comparison, starch-based films were prepared using commercial calcium carbonate, instead of microalgae biomass. As control, pristine starch films (without microalgae biomass or calcium carbonate) were also prepared. All the films were kept for at least 5 days at room temperature under controlled relative humidity (~53%) until further analysis. The procedures used for the production and characterisation of starch-based films are summarized in Figure 1.

### 2.5. Starch-Based Films Characterisation

#### 2.5.1. Optical Properties

A colorimeter (Minolta, Kyoto, Japan) was used to measure CIELab colour parameters: *L**, with values ranging from 0 (black) to 100 (white), related to darkness or lightness; *a**, with negative values related with green and positive values related with red; and *b**, with negative values related with blue and positive values related with yellow. For each film (on a sheet of white paper), five different points were analysed. Moreover, as seen previously [12], the colour variation (ΔE) between starch-based films containing a filler (microalgae biomass or commercial CaCO_3_) and control was calculated as follows:(1)$$\Delta E=\sqrt{{\left(\Delta L^{*}\right)}^{2}+{\left(\Delta a^{*}\right)}^{2}+{\left(\Delta b^{*}\right)}^{2}}$$

#### 2.5.2. Thermal Properties

Thermogravimetric analysis (TGA) was carried out using a simultaneous thermogravimetric analyser (STA300, Hitachi, Tokyo, Japan). Samples (5–10 mg) of selected films (control and those containing 20% of filler) were heated until 600 °C at a heating rate of 10 °C min^−1^ under air atmosphere.

#### 2.5.3. Mechanical Properties

All films were cut in strips (9 cm length and 1 cm width) for determination of tensile properties based on ASTM D 882-2018 standard method and using texture analyser TA.HDi (Stable Micro Systems) equipped with fixed grips lined with a thin rubber.

Film thickness (±0.001 mm accuracy) was determined using a digital micrometre (Mitutoyo Corporation, Japan) and measuring at least five points along the length of each film strip immediately before the tests. Each strip (at least five per film condition) was placed between the grips, leaving 5 cm^2^ as exposed area. The crosshead speed was set at a constant rate of 0.5 mm s^−1^. As in previous studies [34,35], Young’s modulus, tensile strength, and elongation at break were determined from stress–strain curves.

#### 2.5.4. Water Contact Angle

Static water contact angle (WCA) was determined at room temperature on top (exposed to air during solvent evaporation) and down (in direct contact with the acrylic plate during solvent evaporation) surface of each film using a contact angle measuring instrument (OCA 20, Dataphysics). A drop (3 µL) of ultrapure water was dispensed on the surface of each film strip (6 cm length and 1 cm width) with a microsyringe. WCA values were obtained by an image analysis software (Dataphysics SCA20 M4) using the Laplace-Young equation [34,35]. Three film strips of each condition were analysed with at least ten WCA measurements on different points of each film surface.

#### 2.5.5. Moisture Content

Film residual moisture was determined by drying film squares (4 cm^2^) at 105 °C for 16 h into small aluminium foil-shaped boxes (previously dried under the same conditions). After 30 min in a desiccator for cooling, the dried film squares (at least 3 per condition) were weighted, and the moisture percentage was calculated.

#### 2.5.6. Solubility in Aqueous Medium

Film squares (4 cm^2^) were weighed and then immersed in 30 mL of distilled water containing sodium azide (0.02% *w*/*v*) to avoid microbial growth. After 8 days at room temperature under orbital stirring (80 rpm), non-solubilised samples were recovered, dried at 105 °C for 16 h, and weighed after 30 min in a desiccator for cooling. The film solubility was determined (at least in triplicate per film condition) as weight loss percentage.

#### 2.5.7. Water Vapor Permeability

Water vapor permeability (WVP) was determined using the ASTM standard method, as previously described [12]. Anhydrous calcium chloride (pre-dried at 200 °C for 12 h and stabilised for at least 12 h in a desiccator containing silica gel) was placed inside plexiglass permeation cells. Each cell containing desiccant was covered with a circular film specimen (2 cm in diameter) and placed in a chamber with controlled relative humidity (53%) and temperature (25 °C). The permeation cells were weighted at 0, 4, 8, 24, 48, and 72 h. The measurements were carried out in triplicate.

#### 2.5.8. Scanning Electron Microscopy (SEM)

Scanning electron microscopy (SEM) analyses were performed on a SU-70 Hitachi microscope, operating at 4 kV (for films and microalgae) or 15 kV (for commercial CaCO_3_). In the case of microalgae, both fresh culture and freeze-dried biomass were analysed. Each culture was filtered onto a Isopore^TM^ polycarbonate membrane filter (0.4 µm pore size and 37 mm diameter) and rinsed with distilled water. Samples (including those on filters after air-drying) were fixed on SEM specimen holders using double-sided carbon tape. A conductive carbon thin film was then deposited onto the films using a carbon rod coater (Emitech K950X). For selected films (control and those containing 2.5 and 20% of filler), both surfaces were analysed, being denoted Down and Top according to if the surface was in contact or not with the plexiglass plates during the film solvent casting, respectively.

#### 2.5.9. Antioxidant Activity

The antioxidant capacity of the films was determined using ABTS assay [34]. The radical cation ABTS^•+^ was generated by preparing 7 mM ABTS solution in 2.45 mM potassium persulfate, kept in the dark at room temperature for 16 h under stirring. ABTS^•+^ solution was then diluted to obtain a working solution with an absorbance value at 734 nm of about 0.8. A film square (1 cm^2^) was placed in 3 mL of the working solution and kept in the dark under stirring (80 rpm) between absorbance measurements. Absorbance was measured using a microplate spectrophotometer (BioTek Instruments, Winooski, VT, USA) after 30 min of the incubation and then hourly for up 8 h and daily for up to 6 days, recovering the solution in each well after measurement. Triplicates for each film condition were prepared, together with blanks (without a film square). The inhibition percentage of ABTS radical was calculated as follows:(2)$$\mathrm{Inhibition}\;\left(\%\right)=\left(\frac{\mathrm{Absorbance}\;\mathrm{Blank}\;-\;\mathrm{Absorbance}\;\mathrm{Film}}{\mathrm{Absorbance}\;\mathrm{Blank}}\right)\times 100$$

### 2.6. Statistical Analysis

The statistical analysis was carried out in Python [36]. The obtained results from colour, tensile mechanical properties, water contact angle, solubility, water vapour permeability, and antioxidant activity were analysed by F-test, followed by Student’s *t*-test with a significance level of 95%.

## 3. Results and Discussion

The freeze-dried biomass of *E. huxleyi* (EHUX) and *C. pseudoroscoffensis* (CP) used in the production of starch-based films was characterised by elemental analysis and protein estimation (Appendix A). The protein content (16.1 ± 0.1% for EHUX and 11.2 ± 0.1% for CP) was similar to those previously obtained from other batches produced under the same conditions [22,23], suggesting a consistent composition between different cultures. The microalgae under study were also observed by scanning electron microscopy (SEM). Coccoliths and coccospheres were visible with the characteristic morphology described for *E. huxleyi* [25,37] and *Chrysotila* genera [24,38,39]. Of note, among other structural details that are unique to each species, *C. pseudoroscoffensis* coccospheres have higher diameter than *E. huxleyi* coccospheres (ca. 12 µm versus 5 µm) (Figure 1a,b).

Considering their distinct chemical composition and morphology, starch-based films incorporating *E. huxleyi* and *C. pseudoroscoffensis* were produced. As coccolithophores are rich in CaCO_3_, commercial CaCO_3_, having a distinct morphology (Figure 1c), was also used for comparison to prepare starch films.

### 3.1. Characterisation of Starch-Based Films with Microalgae Biomass

#### 3.1.1. Colour and Morphology

The starch-based films containing microalgae biomass, especially for 10% and 20% of *C. pseudoroscoffensis* (CP), presented a green-yellowish hue. Moreover, the incorporation of CaCO_3_ at 10% and 20% resulted in whitish films. Despite that, all the films were transparent, having a suitable appearance for food packaging (real images in Appendix A).

Table 1 shows CIElab parameters and total colour variation for each filler incorporation (statistical analysis results in Appendix A). The commercial CaCO_3_ did not significantly change any colour parameters, except the red-green coordinate (*a** value) for CaCO_3_ 2.5% and the yellow-blue coordinate (*b** value) for CaCO_3_ 10%. For both microalgae (EHUX and CP), luminosity (*L**) and red-green coordinates (*a** value) decreased with increasing biomass content, whereas an increase of yellow-blue (*b**) values were observed. The *L** values decreased from 90.52 ± 0.34 (control) to 84.19 ± 0.25 (Starch + EHUX 20%) and 77.54 ± 0.39 (Starch + CP 20%), and *a** values from 2.05 ± 0.03 (control) to −0.98 ± 0.23 (Starch + EHUX 20%) and −0.05 ± 0.03 (Starch + CP 20%). The *b** values increased from −3.49 ± 0.05 (control) to 11.44 ± 0.26 (Starch + EHUX 20%) and 12.40 ± 0.17 (Starch + CP 20%). As visually observable (Appendix A), the highest microalgae biomass percentage yielded films with less transparency and the highest green-yellowish colouration. Based on the values obtained in the CIELab parameters, the films’ total colour variation (ΔE) was estimated. Such variation increased with the increasing amount of microalgae biomass incorporated, reaching 16.50 ± 0.33 and 20.62 ± 0.37 with 20% of EHUX and CP, respectively. This revealed an influence of microalgae in the optical properties of the starch-based films.

To access the differences on surface morphology, films containing the lowest (2.5%) and highest (20%) amount of each filler, as well as control starch films were observed by SEM on both surfaces, Top (exposed to air during solvent evaporation) and Down (in direct contact with the acrylic plate during solvent evaporation) (Figure 2). Few white dots were visible on both surfaces of the control films, possibly due to some starch not being well-dispersed. In the case of the films prepared by incorporating commercial CaCO_3_ or microalgae, filler was well-immersed in the starch matrix and reasonably well-dispersed. Intact coccoliths were observed in both surfaces (Top and Down). However, some agglomeration was visible, especially for films containing 20% of microalgae biomass.

#### 3.1.2. Thermal Properties

The thermal stability of selected films (control and those containing 20% of filler) was assessed by thermogravimetric analysis (TGA). For all tested films, the TGA curves and first derivative (Appendix A) showed two main stages of weight loss. The first weight loss, with temperature at maximum degradation (T_max_) ranging between 57.6 (for Starch + CP 20%) and 91.1 °C (for Starch + EHUX 20%), was attributed to the loss of moisture in the films. The second weight loss occurred approximately between 225–350 °C. For each film, T_max_ and weight of remaining ash (%) for the second stage of thermal decomposition are summarized in Appendix A. While T_max_ at the second stage decreased with incorporation of microalgae of 295.8 °C (control) to 278.8 °C (CP) and 251.4 (EHUX), it was increased to 302.7 °C by incorporation of commercial CaCO_3_. These results indicate that the thermal stability of the films was reduced with the incorporation of microalgae but increased with commercial CaCO_3_. This reduction is due to the presence of organic compounds as components of microalgae. Accordingly, standards of carbohydrates, lipids, and proteins started to degrade from 150 °C [40]. Although CaCO_3_ decomposition is expected above 600 °C, structural alterations were observed when coccoliths of *E. huxleyi* were treated at 300 °C [41].

The remaining ash increased from 8.0% (control) to 19.7%, 18.6%, and 12.5% in films containing 20% of commercial CaCO_3_, EHUX, and CP, respectively. This corroborates the thermal stability of CaCO_3_, either from commercial CaCO_3_ sample or coccoliths of microalgae. The remaining ash in control was similar to that was previously found in films with tapioca starch films (7.9%) [42].

Despite the reduced thermal stability due to the incorporation of microalgae, the temperature in which these films start to degrade (>225 °C) is compatible with materials processing and diverse applications in food packaging.

#### 3.1.3. Mechanical Properties

The thickness (Figure 3a) of pristine starch films (control) was 28.8 ± 3.4 µm. This value was about 2.5-fold lower than that obtained from films prepared using mixtures containing 4% of starch [12]. However, in this work, a lower content of starch was used (1.5%), which is proportional to the thickness difference. After incorporation of the fillers under study, the thickness of the film ranged from 27.2 ± 1.6 µm (for Starch + EHUX 2.5%) to 44.7 ± 2.9 µm (for Starch + CP 20%). In the case of commercial CaCO_3_ or *E. huxleyi* (EHUX), no significant differences were observed in the thickness of the film for 2.5 and 5% when compared with control, while an increase of thickness was observed for 10 and 20%. The incorporation of *C. pseudoroscoffensis* (CP) biomass significantly increased film thickness for all the filler percentages (2.5, 5, 10, and 20%). The most evident difference between the two microalgae is the larger size of *C. pseudoroscoffensis* coccospheres compared to those of *E. huxleyi* (Figure 1a,b), which can explain the highest thickness observed in the films.

Either with the incorporation of microalgae or commercial CaCO_3_, a decrease in the Young’s modulus (Figure 3b) was observed compared to the control. This reduction was greatest with 20% of filler, decreasing the Young’s modulus from 1361.6 ± 122.8 MPa (control) to 424.4 ± 117.6 MPa for CaCO_3_ 20%. An even a higher decrease was observed for the microalgae biomass incorporation, namely 290.2 ± 62.8 MPa for EHUX 20% and 333.4 ± 156.8 MPa for CP 20%. The decrease in the Young’s modulus indicates that all the fillers turned the films significantly less rigid, suggesting a possible interference with the starch matrix crystallisation. The incorporation of the filler in the starch matrix may reduce interactions between starch chains [12]. The same trend, i.e., reduction of the Young’s modulus, was observed by the addition of microalgae biomass, *Heterochlorella luteoviridis* (0.5, 1 and 2%) or *Dunaliella tertiolecta* (1 and 2%), on cassava starch films [20].

The tensile strength was also significantly reduced with the incorporation of microalgae biomass or commercial CaCO_3_, especially for 20%, when compared to pristine starch films (Figure 3c). Concerning this filler percentage, tensile strength decreased from 28.4 ± 3.9 MPa (control) to 6.7 ± 2.1 MPa (Starch + CaCO_3_ 20%), 2.9 ± 0.9 MPa (Starch + EHUX 20%), and 4.3 ± 2.5 MPa (Starch + CP 20%). A reduction in tensile strength, indicative of less mechanically resistant films, was also observed for corn starch films containing 4% of *N. gaditana* [19], as well as cassava starch films containing 0.5, 1, and 2% of *H. luteoviridis* or *D. tertiolecta* [20].

The elongation at break (Figure 3d) was significantly diminished with incorporation of *C. pseudoroscoffensis* at 10 and 20% and *E. huxleyi* at 20%, decreasing from 2.8 ± 0.5% (control) to 2.0 ± 0.3%, 1.9 ± 0.3%, and 1.4 ± 0.2%, respectively. This decrease indicates a significant decrease in the film’s extensibility.

#### 3.1.4. Wettability, Water Solubility, and Water Vapour Permeability

The water contact angle (WCA), an indicator of the material hydrophilic (<90°) or hydrophobic (≥90°) character, was measured on both film surfaces (Top and Down) (Figure 4). The pristine starch films (control) showed a WCA of 55.5° ± 4.4° on Top and 50.7° ± 5.7° on Down, indicating their hydrophilicity. All the films containing CaCO_3_ showed a higher WCA on the top surface than on the down surface. Considering this filler, the highest WCA on top (94.8 ± 12.5°) and down (72.0 ± 6.5°) surfaces were obtained for films containing 2.5% of CaCO_3_. When compared with respective surfaces of the control film, the incorporation of microalgae biomass (EHUX or CP) at 2.5%, 5%, and 10% increased the WCA of both top and down surfaces, with average values ranging from 103.8° ± 7.1° (Starch + CP 10%; down) to 121.3° ± 6.6° (Starch + CP 5%; top). Similarly, an increase was observed in WCA of the top surface of starch-based films containing 20% of EHUX (112.9° ± 6.7°) or CP (107.3° ± 9.4°), revealing a hydrophobic character. However, this trend was not observed for the respective down surfaces, whose WCA values were not significantly changed (in the case of Starch + EHUX 20%) or decreased (in the case of Starch + CP 20%) relative to the control.

The incorporation of microalgae hydrophobic compounds, such as lipids and pigments found in the coccolithophores under study [22,23,24], may explain the hydrophobic nature of the films (WCA > 90°). The increase observed in water tolerance on the top surface, but not on the down, of the films containing 20% of microalgae biomass may be due to the separation of compounds with lower water solubility (e.g., lipids) and density during solvent evaporation. These compounds can be concentrated on the top surface, originating a heterogeneity in the films surfaces that affects the wetting properties [35]. In another study, the incorporation of *N. gaditana* biomass at 4% turned corn starch-based films more hydrophobic, but the films retained a hydrophilic behaviour (WCA of 48.8°) [19]. This difference compared to the present work may be related to a distinct chemical composition of the microalgae. The increase of surface hydrophobicity led to a decrease in water absorption by the films from the food moisture, helping to improve the barrier properties of hydrophilic films, such as starch ones.

The solubility of the starch films (control) and films containing 10% or 20% of filler was determined by immersion in water for 8 days (Appendix A). The weight loss of the control was 22.9 ± 5.2%. A weight loss of about 30% was previously obtained with pristine starch films immersed in water for 7 days [12,35]. This loss was related to the weak interaction of the glycerol with the starch network, which is released by diffusion to the water [12,35]. In this study, no significant differences were found for films containing 10% or 20% of filler compared to control, except for Starch + EHUX 10% films and Starch + CP 20% films that lost 12.6 ± 7.0% and 34.2 ± 2.3% of their weight, respectively. The higher solubility of Starch + CP compared to Starch + EHUX films may be related to the higher content of hydrophilic compounds in microalgae biomass, as lipids only represented 6.4% in *C. pseudoroscoffensis* [23], whereas *E. huxleyi* had 20.3% [22].

Figure 5 shows the effect of the incorporation of commercial CaCO_3_ or microalgae biomass on water vapour permeability (WVP) of starch-based films (determined after 48 h in permeation cells containing desiccant). Films used as control showed a WVP of 83.2 ± 8.7 pg Pa^−1^ s^−1^ m^−1^. This value was about two times higher than that obtained with films prepared using starch-recovered potato washing slurries [12]. Still, these films had proportionally greater thickness (about 70 µm) [12], which is a characteristic that greatly influences permeability. The filler incorporation generally did not significantly change WVP values compared to the control, with only three exceptions. A significant decrease was observed using CaCO_3_ at 2.5% (WVP of 57.7 ± 4.5 pg Pa^−1^ s^−1^ m^−1^) and EHUX at 5% (51.5 ± 8.1 pg Pa^−1^ s^−1^ m^−1^), while the incorporation of CP at 10% increased WVP to 107.8 ± 4.4 pg Pa^−1^ s^−1^ m^−1^. A film with an effective water barrier property is required to prevent undesirable changes in food, mainly for the packaging of dry foods.

#### 3.1.5. Antioxidant Activity

The ABTS^•+^ inhibition (determined after 5 h of incubation; Figure 6) increased from 0.9% ± 0.2% (control) to 6.0% ± 2.1%, 11.6% ± 3.4%, 6.9% ± 2.0%, and 13.0% ± 2.2% with the incorporation of 2.5, 5, 10, and 20% of CaCO_3_, respectively. The use of microalgae biomass (EHUX or CP) led to a greater increase in the ABTS^•+^ inhibition than that observed for CaCO_3_. The ABTS^•+^ inhibition of Starch + EHUX films ranged between 11.1% ± 1.1% and 60.4% ± 5.6%, whereas Starch + CP films ranged between 10.7% ± 2.4% and 44.8% ± 4.8%. For both microalgae under study, the antioxidant activity increased by increasing the amount of biomass incorporated. Similarly, the addition of *Tetradesmus obliquus* biomass improved the antioxidant activity of corn starch films (plasticised with glycerol and blended with polyallylamine), and the antioxidant potential was increased by increasing the microalga content [21]. The bioactive constituents of microalgae confer antioxidant properties to the films. Indeed, polar lipids and pigments with antioxidant potential were previously identified in the coccolithophore species used in this work [22,23,24]. The antioxidant capacity could prevent the food degradation by oxidation of some susceptible compounds.

## 4. Conclusions

In this study, starch-based films containing microalgae biomass (*E. huxleyi* or *C. pseudoroscoffensis*) were produced and compared to films produced under the same conditions but using commercial calcium carbonate as filler. Both microalgae revealed to be suitable for developing green-yellowish transparent films with hydrophobic nature and high antioxidant activity, which are required properties for food packaging materials, mainly to increase food shelf-life by preventing oxidation. These properties were not observed for the whitish transparent films that incorporated commercial calcium carbonate. Hence, this work proved that coccolithophore microalgae, independently of the species, have promising bioactive compounds that can be valued in the formulation of sustainable active food packaging. The results pave the way for the direct use of algae biomass in bioplastic, which can be included by extrusion processing in the thermoplastic starch matrix for industrial transposition. Further research and development are needed, particularly to find the best applications in food packaging.

## Data Availability

Data is contained within the article.

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
