# Peer review of "Potential of Coccolithophore Microalgae as Fillers in Starch-Based Films for Active and Sustainable Food Packaging"

_foods, 2023, doi:10.3390/foods12030513_

Round 1

Reviewer 1 Report

In this study, Coccolithophore microalgae, such as Emiliania huxleyi (EHUX) and Chrysotila pseudoroscof- 22 fensis (CP), which are composed of calcium carbonate and contain bioactive compounds, were incorporated as fillers in starch-based films, to develop sustainable biodegradable and bioactive materials for food packaging applications. The study is well written and can be of interest for material scientists dealing with packaging materials. However, this research is focused on analyzing the film properties (mechanical, barrier, morphological, antioxidant properties, etc.) and it does not include any study of food shelf life or even involve the use of food residue. Therefore, in my opinion, this is not within the scope of the journal Foods.

Moreover, the following minor comments can be addressed:

-     - Equations in section 2.4.2, 2.4.4, 2.4.5 and 2.4.6 are very general and can be removed

-       - Real images of the film samples can show the whiting effect of the fillers

-     - The oxygen permeability values can also be of interest to ascertain the effect of the fillers on the starch matrix

Author Response

Comment 1: In this study, Coccolithophore microalgae, such as Emiliania huxleyi (EHUX) and Chrysotila pseudoroscoffensis (CP), which are composed of calcium carbonate and contain bioactive compounds, were incorporated as fillers in starch-based films, to develop sustainable biodegradable and bioactive materials for food packaging applications. The study is well written and can be of interest for material scientists dealing with packaging materials. However, this research is focused on analyzing the film properties (mechanical, barrier, morphological, antioxidant properties, etc.) and it does not include any study of food shelf life or even involve the use of food residue. Therefore, in my opinion, this is not within the scope of the journal Foods.

Authors answer: We thank the Reviewer for the positive comment about our paper. Indeed, it is a first study on the incorporation of coccolithophore microalgae in starch-based films. These films have the required properties for food applications due to the barrier and antioxidant properties. Furthermore, this paper is included in a special issue named “The Application of Microalgae for the Development of High-Added-Value Products”. As the main aim of the paper it is the development of starch-based bioplastics with improved properties due to the incorporation of the microalgae biomass, we think that it is in the scope of the issue and also the journal. Indeed, in future we think that will be interesting to evaluate the impact of this material on food shelf life but, at the moment this is not possible, since there are some limitations related to the amount of microalgae biomass available, as the cultivation conditions are still being established for industrial scale up.

Comment 2: Moreover, the following minor comments can be addressed:

- Equations in section 2.4.2, 2.4.4, 2.4.5 and 2.4.6 are very general and can be removed

- Real images of the film samples can show the whiting effect of the fillers

- The oxygen permeability values can also be of interest to ascertain the effect of the fillers on the starch matrix

Authors answer: Considering the Reviewer comment, Equations in section 2.4.2, 2.4.4, 2.4.5 and 2.4.6 were removed. Real images of the film samples are shown in Supplementary Figure S1. We agree that oxygen permeability values can also be of interest, but at the moment we could not do this analysis due to algal biomass limitations.

Reviewer 2 Report

Biomass are promising materials for the design of active and sustainable packaging. The topic of this manuscript is of interest to the readers and the experiments are well designed. However, major revision is required.

1.     Which kind of food does the designed films aim to?

2.     Renewable sources are promising precursors for biodegradable active packaging. Many researches have been done and more references are suggested to be cited for broad readers, for example Development and Characterization of Food Packaging Bioplastic Film from Cocoa Pod Husk Cellulose Incorporated with Sugarcane Bagasse Fibre; Packaging and degradability properties of polyvinyl alcohol/gelatin nanocomposite films filled water hyacinth cellulose nanocrystals; Electrospun Functional Materials toward Food Packaging Applications: A Review.

3.     The authors should mention where did they get the commercial CaCO3.

4.     Please pay attention to the writing of subscripts and superscripts, e.g. “cm2” in line 173 need to be revised. Please double check the manuscript to remove typos.

5.     “µmol m-2 s-1”, “mm/s” and other units should be written in the same way.

6.     How about the thermal stability of composite films?

7.     How long does the composite films could be degraded naturally?

Author Response

Comment 1: Biomass are promising materials for the design of active and sustainable packaging. The topic of this manuscript is of interest to the readers and the experiments are well designed. However, major revision is required.

Authors answer: We thank the Reviewer for the positive appreciation of the manuscript, as well as for the other comments addressed point-by-point below.

Comment 2: Which kind of food does the designed films aim to?

Authors answer: Considering that the incorporation of microalgae biomass improved hydrophobicity and antioxidant capacity of starch-based films, different applications in food packaging could be considered. As example, these films could be tested to pack smoked fish fillets, as previously done for other starch-based films. In addition, the packaging of dry food is also envisaged for this type of material. This is a future step, following the optimization of the cultivation conditions to produce microalgae biomass on an industrial scale.

Comment 3: Renewable sources are promising precursors for biodegradable active packaging. Many researches have been done and more references are suggested to be cited for broad readers, for example Development and Characterization of Food Packaging Bioplastic Film from Cocoa Pod Husk Cellulose Incorporated with Sugarcane Bagasse Fibre; Packaging and degradability properties of polyvinyl alcohol/gelatin nanocomposite films filled water hyacinth cellulose nanocrystals; Electrospun Functional Materials toward Food Packaging Applications: A Review.

Authors answer: Indeed, considering the work developed, we focused the introduction on starch-based films, especially those obtained by solvent casting. However, as suggested by the Reviewer, new references were added for broad readers:

Azmin, S.N.H.M.; Hayat, N.A.b.M.; Nor, M.S.M. Development and characterization of food packaging bioplastic film from cocoa pod husk cellulose incorporated with sugarcane bagasse fibre. J. Bioresour. Bioprod. 2020, 5, 248-255.

Oyeoka, H.C.; Ewulonu, C.M.; Nwuzor, I.C.; Obele, C.M.; Nwabanne, J.T. Packaging and degradability properties of polyvinyl alcohol/gelatin nanocomposite films filled water hyacinth cellulose nanocrystals. J. Bioresour. Bioprod. 2021, 6, 168-185.

Mohamed, S.A.A.; El-Sakhawy, M.; El-Sakhawy, M.A.-M. Polysaccharides, protein and lipid -based natural edible films in food packaging: a review. Carbohydr. Polym. 2020, 238, 116178.

Sid, S.; Mor, R.S.; Kishore, A.; Sharanagat, V.S. Bio-sourced polymers as alternatives to conventional food packaging materials: A review. Trends Food Sci. Technol. 2021, 115, 87-104.

Comment 4: The authors should mention where did they get the commercial CaCO3.

Authors answer: As indicated in Section 2.1, commercial calcium carbonate was purchased from Merck (Germany).

Comment 5: Please pay attention to the writing of subscripts and superscripts, e.g. “cm2” in line 173 need to be revised. Please double check the manuscript to remove typos. “µmol m-2 s-1”, “mm/s” and other units should be written in the same way.

Authors answer: We thank the comment of the Reviewer. As suggested, the entire manuscript was revised to remove typos.

Comment 6: How about the thermal stability of composite films?

Authors answer: Considering the comment of the Reviewer, new data on thermal stability of the films were acquired. Thermogravimetric analysis data were included into the manuscript (new sections 2.4.2 and 3.1.2) and supplementary material (Table S2 and Figure S3).

Comment 7: How long does the composite films could be degraded naturally?

Authors answer: Indeed, the biodegradability of the films was not assessed. The degradation period described in the literature for starch-based films in soil is quite variable, namely ranging from 5 days (de Azevedo, L.C.; Rovani, S.; Santos, J.J.; Dias, D.B.; Nascimento, S.S.; Oliveira, F.F.; Silva, L.G.A.; Fungaro, D.A. Biodegradable Films Derived from Corn and Potato Starch and Study of the Effect of Silicate Extracted from Sugarcane Waste Ash. ACS Applied Polymer Materials 2020, 2, 2160-2169) to 56 days (Luchese, C.L.; Benelli, P.; Spada, J.C.; Tessaro, I.C. Impact of the starch source on the physicochemical properties and biodegradability of different starch-based films. J. Appl. Polym. Sci. 2018, 135, 46564).

Reviewer 3 Report

This manuscript is well designed and written, the objectives are clear, and the topic is worthy of investigation. Please find some  suggestions for correction:

1. The title seems to be not accurate enough, in which starch-based films were not specified. Additionally, their potential application in food packaging was not evaluated in this study.

2. All equations should be numbered and expressed in a clearer way.

3. Line 168, the standard method ASTM D 882-83 can be updated to ASTM D 882-2018.

4. Pay attention to the subscript, such as CaCO3.

5. Thermal Properties of the developed films should be assessed by thermogravimetric analysis (TGA) and/or differential scanning calorimetry (DSC).

Author Response

Comment 1: This manuscript is well designed and written, the objectives are clear, and the topic is worthy of investigation. Please find some suggestions for correction:

Authors answer: We thank all the comments and suggestions.

Comment 2: The title seems to be not accurate enough, in which starch-based films were not specified. Additionally, their potential application in food packaging was not evaluated in this study.

Authors answer: Considering the comment of the Reviewer, the tittle was changed for the following: “Potential of coccolithophore microalgae as fillers in starch-based films for active and sustainable food packaging”. We decided to maintain the reference to food packaging, since this material was developed for this application due to its properties, although the impact of this material on food shelf life was not evaluated.

Comment 3: All equations should be numbered and expressed in a clearer way.

Authors answer: We decided to remove the equations since they are general and described in other papers, as suggested by one reviewer.

Comment 4: Line 168, the standard method ASTM D 882-83 can be updated to ASTM D 882-2018.

Authors answer: The correction was done, as indicated by the Reviewer.

Comment 5: Pay attention to the subscript, such as CaCO3.

Authors answer: As suggested, subscripts were checked.

Comment 6: Thermal Properties of the developed films should be assessed by thermogravimetric analysis (TGA) and/or differential scanning calorimetry (DSC).

Authors answer: As mentioned in the answer to the Reviewer 2 Comment 6, new data on the thermal properties (assessed by thermogravimetric analysis) were included into the manuscript.

Reviewer 4 Report

I have carefully read the manuscript entitled “Potential of coccolithophore microalgae as fillers for active and sustainable food packaging” by Moreira et al. I found this topic interesting but I have few concerns related to the research article. I am asking authors to revise the manuscript carefully considering my comments. I have given my comments so that authors to rethink and improve the quality of the manuscript:

Abstract:

The main feature of the research study and how it can impact future research should be highlighted in the abstract.

Quantitative data related to main finding must be mentioned in the abstract. Authors just mentioned increase or decrease of expressions. This need to be improved.

Introduction:

Authors must give some of the information regarding coccolithophore microalgae initially rather than later part of introduction and how this algae is economical important and give some information about safety aspects utilizing this algae in active food packaging.

Authors should mention how the scope of the current study is different from the previous studies in last para of introduction.

I have found the manuscript with self-duplications. Some of the part mentioned in introduction is overlapping with that of discussion part. I suggest to restructure the manuscript so that self -duplication can be avoided.

Material and methods:

Material, chemical and other components used are not mentioned in the material methodology. I must be mentioned separately in a section 2.1. Materials (make should be mentioned).

Authors must add a flow diagram showing methodology followed in the experimentation and also show the analysis performed at each stage. This flow diagram will definitely improve the readability of the manuscript.

Result and Discussion:

Authors have presented their finding in well manner but related discussion is not sufficient (even I can say no discussion from previous studies). It is suggested to improve the manuscript considering this specific aspect. This is most weakest part of the manuscript.

I wish to see a revised version of the manuscript with a good discussion throughout the result and discussion section with recent references.

Way forward to future research must be mentioned in the conclusion.

Tables and figures must be supplemented with appropriate statistical description (wherever applicable).

Moreover, English language needs moderate improvements. Punctuations and spacing also need to be checked carefully throughout the manuscript.

If authors can take serious efforts to improve the manuscript, I will be happy to re-review the manuscript.

Author Response

Comment 1: I have carefully read the manuscript entitled “Potential of coccolithophore microalgae as fillers for active and sustainable food packaging” by Moreira et al. I found this topic interesting but I have few concerns related to the research article. I am asking authors to revise the manuscript carefully considering my comments. I have given my comments so that authors to rethink and improve the quality of the manuscript.

Authors answer: We thank the reviewer for the careful read of our manuscript, as well as all comments that we considered during the revision to improve our paper.

Comment 2: Abstract: The main feature of the research study and how it can impact future research should be highlighted in the abstract.

Authors answer: This study is the first one using coccolithophore microalgae in the formulation of starch-based films, envisioning the development of sustainable biodegradable and bioactive materials for food packaging applications. The findings here achieved are a basis for further research and development. Considering the comment of the Reviewer, these points were highlighted in the abstract, with the following sentences:

“In this study, for the first time these microalgae were incorporated as fillers in starch-based films, envisioning the development of sustainable biodegradable and bioactive materials for food packaging applications.”

“These findings should be considered for further research using coccolithophores to produce active and sustainable food packaging material.”

Comment 3: Abstract: Quantitative data related to main finding must be mentioned in the abstract. Authors just mentioned increase or decrease of expressions. This need to be improved.

Authors answer: We have some limitations regarding the maximum number of words. However, as suggested by the Reviewer, quantitative data were added in the abstract. Accordingly, three sentences were rephrased, as follows:

The incorporation of CaCO3 and microalgae biomass (EHUX or CP) made the films significantly less rigid, decreasing Young’s modulus up to 4.7-fold.

Also, the incorporation of microalgae hydrophobic compounds as lipids turned the surface hydrophobic (water contact angles > 90°).

Contrarily to what was observed with commercial CaCO3, the films prepared with microalgae exhibited antioxidant activity, increasing from 0.9% (control) up to 60.4% (EHUX 20%) of ABTS radical inhibition.

Comment 4: Introduction: Authors must give some of the information regarding coccolithophore microalgae initially rather than later part of introduction and how this algae is economical important and give some information about safety aspects utilizing this algae in active food packaging.

Authors answer: To the best of our knowledge, coccolithophore microalgae E. huxleyi and C. pseudoroscoffensis are not yet produced on an industrial scale. The scale up of the cultivation to industrial level is still needed to boost their potential applications and economical value. Considering the comment of the Reviewer, the introduction was completed focusing this point and additional information regarding coccolithophore microalgae. The new sentences are as follows:

Most studies on coccolithophores have been performed with Emiliania huxleyi (an abundant and widely distributed species in the oceans) and focused their role on the global ocean calcification and carbon cycle (e.g. [29]), or their unique coccolith structures (e.g. [26]). Indeed, little focus have been done on the potential of coccolithophores as a raw material, or source of selected valuable compounds, for development of new products and materials. Such studies, along with the sustainable cultivation of these microalgae on an industrial scale to ensure biomass stocks with a constant composition for commercial uses, are needed to boost the economic value of coccolithophore microalgae.

Regarding the chemical composition of microalgae, it is well-known that it depends on the grown conditions (e.g. light, temperature and nutrients), but it is also species-specific [13]. Previous study with cultured E. huxleyi biomass revealed 50.8% ash, 20.3% lipids, 15.3% proteins, and 3.7% sugars [22]. Using the same methodologies for characterisation of other coccolithophore species, Chrysotila pseudoroscoffensis (originally named as Pleurochrysis pseudoroscoffensis) showed 45.5% ash, 11.6% proteins, 11.0% carbohydrates, and 6.4% lipids [23].

Considering their distinct chemical composition, in this study E. huxleyi and C. pseudoroscoffensis biomasses were incorporated as fillers in starch-based films, envisioning the development of biodegradable and bioactive materials for food packaging applications. The present work is the first one using coccolithophore microalgae for such purpose, particularly in the development of starch-based films.   The films were produced by solvent casting, and the microalgae biomass effect on the morphological, mechanical, barrier, and antioxidant properties was evaluated. Commercial CaCO3(calcite), a widely used filler in the plastic industry [30], was used for comparison purposes.

Comment 5: Introduction: Authors should mention how the scope of the current study is different from the previous studies in last para of introduction.

Authors answer: As mentioned in answer to Reviewer 4 Comment 2, this study is the first using coccolithophore microalgae in the formulation of starch-based films. Considering the comment of the Reviewer, a new sentence was added, as follows: “The present work is the first one using coccolithophore microalgae for such purpose, particularly in the development of starch-based films.”

Comment 6: Introduction: I have found the manuscript with self-duplications. Some of the part mentioned in introduction is overlapping with that of discussion part. I suggest to restructure the manuscript so that self -duplication can be avoided.

Authors answer: As suggested, the manuscript was revised, and sentences were restructured to avoid self-duplication.

Comment 7: Material and methods: Material, chemical and other components used are not mentioned in the material methodology. I must be mentioned separately in a section 2.1. Materials (make should be mentioned).

Authors answer: Material, chemical and other components used were presented in each subsection of “Material and methods” section. As suggested, a new section 2.1 “Materials” was created and the following text was added: “Potato starch and anhydrous calcium chloride were purchased from Sigma Aldrich (St Louis, MO, USA). Glycerol was purchased from Fisher Chemicals (98%), calcium car-bonate from Merck (Germany) and 2,2′-azino-bis(3-ethylbenzothiazoline-6-sulfonic acid) (ABTS) from Fluka (USA). Sodium azide was distributed from Panreac Quimica SAU (Barcelona, Spain). All the used reagents were of analytical grade.”

Comment 8: Material and methods: Authors must add a flow diagram showing methodology followed in the experimentation and also show the analysis performed at each stage. This flow diagram will definitely improve the readability of the manuscript.

Authors answer: Considering the comment of the Reviewer, a flow diagram was created and included in “Materials and methods” section (Scheme 1).

Comment 9: Result and Discussion: Authors have presented their finding in well manner but related discussion is not sufficient (even I can say no discussion from previous studies). It is suggested to improve the manuscript considering this specific aspect. This is most weakest part of the manuscript. I wish to see a revised version of the manuscript with a good discussion throughout the result and discussion section with recent references.

Authors answer: Indeed, there are few studies exploring the incorporation of microalgae or CaCO3 in starch-based films obtained by solvent casting. We try to discuss with similar materials developed in the literature, namely using microalgae biomass and CaCO3 (e.g. “The same trend, i.e. reduction of the Young’s modulus, was observed by the addition of microalgae biomass, Heterochlorella luteoviridis (0.5, 1 and 2%) or Dunaliella tertiolecta (1 and 2%), on cassava starch films [17]” or “In another study, the incorporation of N. gaditana biomass at 4% turned corn starch-based films more hydrophobic, but the films retained a hydrophilic behaviour (WCA of 48.8°) [16].”). For this reason, it is not possible to have a very exhaustive comparison with other works. We always explain all the results obtained and compare between all the samples.

Comment 10: Way forward to future research must be mentioned in the conclusion.

Authors answer: As suggested by the Reviewer, a new sentence related to future research was included in the conclusion, as follows: “Further research and development are needed, particularly to find the best applications in food packaging.”

Comment 11: Tables and figures must be supplemented with appropriate statistical description (wherever applicable).

Authors answer: As suggested by the Reviewer, captions of Tables S2-S3 and Figures 3-6 were supplemented with statistical description, as follows. “Different letters between each condition indicate significant differences (Student’s t-test; p < 0.05).”

Comment 12: Moreover, English language needs moderate improvements. Punctuations and spacing also need to be checked carefully throughout the manuscript.

Authors answer: As suggested by the Reviewer, the entire manuscript has been carefully revised to improve English language and check punctuations and spacing.

Comment 13: If authors can take serious efforts to improve the manuscript, I will be happy to re-review the manuscript.

Authors answer: We thank to the Reviewer. All the comments were considered and addressed point-by-point in the revision.

Round 2

Reviewer 1 Report

It can be accepted for publication

Reviewer 2 Report

The manuscript has been revised according to the comments and suggest to be accepted.

Reviewer 4 Report

Improved sufficiently. May be accepted for publication.